# Yeast Lsm Pro-Apoptotic Mutants Show Defects in Autophagy

**DOI:** 10.3390/ijms241813708

**Published:** 2023-09-05

**Authors:** Benedetta Caraba, Mariarita Stirpe, Vanessa Palermo, Ugo Vaccher, Michele Maria Bianchi, Claudio Falcone, Cristina Mazzoni

**Affiliations:** Department of Biology and Biotechnologies “C. Darwin”, Sapienza University of Rome, Piazzale Aldo Moro 5, 00185 Roma, Italy; benedetta.caraba@uniroma1.it (B.C.); mariaritastirpe@gmail.com (M.S.); vanessa.palermo76@gmail.com (V.P.); ugo_vaccher@yahoo.it (U.V.); michele.bianchi@uniroma1.it (M.M.B.); claudio.falcone48@gmail.com (C.F.)

**Keywords:** autophagy, ageing, *LSM4*, yeast

## Abstract

*LSM4* is an essential yeast gene encoding a component of different LSM complexes involved in the regulation of mRNA splicing, stability, and translation. In previous papers, we reported that the expression in *S. cerevisiae* of the *K. lactis LSM4* gene lacking the C-terminal Q/N-rich domain in an *Lsm4* null strain *S. cerevisiae* (*Sclsm4Δ1*) restored cell viability. Nevertheless, in this transformed strain, we observed some phenotypes that are typical markers of regulated cell death, reactive oxygen species (ROS), and oxidated RNA accumulation. In this paper, we report that a similar truncation operated in the *S. cerevisiae LSM4* gene confers on cells the same phenotypes observed with the *K. lactis lsm4Δ1* gene. Up until now, there was no evidence of the direct involvement of *LSM4* in autophagy. Here we found that the *Sclsm4Δ1* mutant showed a block in the autophagic process and was very sensitive to nitrogen starvation or treatment with low doses of rapamycin, an inducer of autophagy. Moreover, both during nitrogen starvation and aging, the *Sclsm4Δ1* mutant accumulated cytoplasmic autophagy-related structures, suggesting a role of Lsm4 in a later step of the autophagy process.

## 1. Introduction

The LSM (like-Sm) protein family is a group of evolutionarily conserved proteins that have been found in a wide range of organisms, from bacteria to yeast and humans [1]. *LSM4* is an essential gene, and it is a part of several LSM complexes that control the stability, splicing, and translation of mRNA [2].

*LSM4* has also been found to be upregulated in several types of cancer, such as breast cancer, and in early-stage pancreatic ductal adenocarcinoma, where it is associated with a poor prognosis [3,4,5].

We previously reported that Lsm4 is involved in aging and apoptosis in the yeast *Saccharomyces cerevisiae*, in which cells expressing a truncated protein of KlLsm4 from the related yeast *Kluyveromyces lactis* (*Kllsm4Δ1*) age prematurely and undergo regulated cell death [6,7]. We showed that *KlLSM4*’s C-terminal Q/N-rich domain is essential for effective RNA degradation [7,8] and P-body localization [9,10], whereas other studies suggested that the *ScLSM4*’s C-terminal Q/N-rich domain is only significant when combined with the Edc3 protein [11]. The use of slightly different constructions and yeast strains with potentially different genetic backgrounds could both account for these variations.

Although the C-terminal domain of Lsm4 in most eukaryotes consists of an arginine–glycine–glycine repeat (RGG) domain rather than a Q/N-rich region, despite a great degree of variation in the primary sequence, some functions seem conserved through evolution, while others were not. It has been reported that low-complexity polypeptide regions of proteins, including R/G-rich regions, were involved in protein polymerization and aggregation, suggesting a role for the Lsm4 C-terminus in these processes [12].

Actually, as observed in yeast, the C-terminal RGG domain of human Lsm4 plays an important role in processing bodies (P-bodies, PB) accumulation, but, differently from yeast, it is not required for the association to the Lsm1–7 complex [13].

This region is crucial for effective histone mRNA degradation in both human and yeast cells [14,15], but it has also been demonstrated that the Lsm4 RGG domain is not a limiting factor for the degradation of histone H2A mRNA [13].

Lsm proteins were found to be involved in the autophagic process by genetic and mRNA capture analyses [16,17,18], and in recent years, it has become clear that mRNA degradation pathways and autophagy are closely related [19].

The amount of autophagy in mutant cells that lack the RNA helicase Dhh1 or the decapping protein Dcp2 increases when nutrients are present. In fact, TORC1 can phosphorylate a serine residue on the Dcp2 that, along with the RCK-Dhh1 complex, binds messenger RNAs of the ATG genes, facilitating the removal of the cap and their degradation by the exoribonuclease Xrn1 when nutrients are present [20]. When nutrients are in short supply or nonexistent, TORC1’s catalytic activity is diminished. As a result, Dcp2’s levels of phosphorylation are diminished, which in turn results in decreased decapping activity and the stabilization of *ATG* gene transcripts. However, it was discovered that the *DHH1* and *DCP2* genes had no effect on the mRNA degradation levels of the *ATG* genes in the absence of nutrients, indicating that the decapping machinery only encourages the degradation of *ATG* transcripts when autophagy is not required [20].

In a more recent study, it was reported that cells lacking the *DHH1* gene (*dhh1*Δ) rapidly lose viability after prolonged nitrogen starvation, indicating that under these conditions there would be a defect in autophagy induction [21].

Therefore, Dhh1 would be a two-way regulator of autophagy: (a) in nutrient-rich conditions, Dhh1 works with the mRNA decapping machinery to degrade ATG mRNAs to maintain autophagy at a basal level; (b) in nitrogen-starved conditions, Dhh1 switches to a facilitative role to encourage the translation of *ATG1* and *ATG13* mRNAs to promote autophagy [21].

A similar bidirectional role in autophagy seems to be played by the CCR4/NOT complex, which, under nutrient-rich conditions, directly targets some *ATG* genes’ mRNAs, promoting their degradation through deadenylation, while upon starvation, CCR4-Not switches its role to promote the expression of a different subset of *ATG* genes required for autophagy induction [22].

Along with the decapping process followed by degradation in the 5’→3’ direction, mRNAs can also be degraded in the 3’→5’ direction by the exosome complex. Although both 5’→3’ and 3’→5’ RNA degradation mechanisms have been extensively studied, little is known about the relationship between the two pathways. The RNA-binding protein Pat1/Mrt1 has been proposed as a possible link between 5’→3’ and 3’→5’ mRNA degradation. Pat1 interacts with the ring-shaped heptameric complex Lsm1-Lsm7 proteins to form the Pat1-Lsm complex that binds to the 3’ untranslated region (UTR) of oligoadenylated mRNA [23,24] and acts as a decapping enhancer, protecting the 3’ end of oligoadenylated mRNA from trimming and exosome-dependent 3’-5’ degradation [25,26,27,28].

The Pat1/Lsm complex preferentially binds to a subset of the *ATG* genes’ mRNA during nitrogen deprivation, preventing the exosome from destroying them and encouraging their accumulation to ensure a strong autophagic response [28]. Indeed, the autophagic process induced by nitrogen starvation exhibits obvious defects in the presence of mutants in the *LSM1* or *PAT1* genes.

The role of the Pat1/Lsm complex in protecting from exosome degradation is specific for some ATGs mRNA, in that the bulk and specific degradation of subsets of mRNAs, notably those encoding amino acid biosynthesis and ribosomal proteins, occurs by nitrogen starvation or rapamycin-induced autophagy in yeast [29,30].

As reported before, the Lsm4 C-terminal Q/N-rich domain is involved in maintaining cell viability during the Chronological Life Span (CLS). In its absence, we observed all typical markers of regulated cell death, together with Reactive Oxygen Species (ROS) and oxidated RNA accumulation [31]. As these phenotypes were observed expressing the heterologous truncated *Kllsm4* gene of *K. lactis* in the deletion mutant of *S. cerevisiae*, we decided to construct the corresponding truncated mutant of the *LSM4* gene of *S. cerevisiae* and perform different analyses to establish the role of the *LSM4* gene in the autophagy process.

## 2. Results

### 2.1. The Sclsm4Δ1 Mutant Shows Regulated Cell Death Markers and Premature Aging

We have previously described how an *S. cerevisiae* strain lacking the *LSM4* gene was able to regain viability by the expression of *Kllsm4Δ1*, a truncated form of the *KlLSM4* gene from the yeast *K. lactis*. Nevertheless, cells lost viability very soon and showed the markers of regulated cell death [6,7]. In order to investigate this phenotype in a homologous scenario, we constructed the corresponding *Kllsm4Δ1* mutant of the *S. cerevisiae LSM4* gene (*Sclsm4Δ1*), and we expressed it in the *S. cerevisiae* MCY4 strain, which contained the *LSM4* gene under the control of the Gal1-10 promoter. This strain can grow on galactose, but it cannot grow when glucose is the only carbon source. The expression in such a strain of the *Sclsm4Δ1* gene from a centromeric plasmid restored growth on glucose, as also reported when *Kllsm4Δ1* was present, although the growth was slightly slower than the wild strain and the one expressing the full form of the *LSM4* gene (Appendix A). To check if the phenotypes of the *Sclsm4Δ1* mutant were similar to those described for *Kllsm4Δ1*, we analyzed nuclei morphology, intracellular ROS production, and maintenance of viability during the stationary phase, also defined as Chronological Life Span (CLS). As shown in Figure 1, in panels A and B, highly fragmented, enlarged, and diffused nuclei, indicative of regulated cell death, were observed in more than 15% of exponentially growing cells and in almost 40% of cells during the stationary phase. These percentages are much higher compared to the wild type, which showed about 3% and 12% of cells with fragmented nuclei in exponential and stationary phase cells, respectively. Similarly to the wild type, the strain expressing the full-length *ScLSM4* gene showed about 1% and 3.5% of cells with fragmented nuclei in the exponential and stationary phases, respectively.

Concerning intracellular ROS accumulation, the percentage of ROS-positive cells during the exponential phase was about 2% and 12% for the WT and the *Sclsm4Δ1* mutant cells, and about 4.7% in the *ScLSM4*-expressing strain (Figure 1, Panels C and D).

One particular phenotype of *Kllsm4Δ1* was the early loss of viability during aging. As shown in Figure 1E, *Sclsm4Δ1* cells also show a very short CLS, as previously demonstrated for *Kllsm4Δ1* [8] (Figure 1, panel E).

As a control, to verify that these phenotypes were not due to the expression of the *Sclsm4Δ1* gene from a centromeric plasmid, we also expressed in the MCY4 strain the full-length gene *ScLSM4*. As shown in Figure 1E, the expression of the *ScLSM4* gene restored the CLS at the same level as the WT strain. To evaluate the ability to respond to starvation conditions, we determined the CLS in SD medium lacking a nitrogen source. As shown in Figure 1F, nitrogen starvation increased CLS in a wild-type strain. On the contrary, MCY4-expressing *Sclsm4Δ1* cells in SD-N showed a drop in viability already after 1 day and completely lost viability at day 4, one day before cells maintained in SD.

Finally, as also reported for *Kllsm4Δ1* [6], *Sclsm4Δ1* showed sensitivity to caffeine and acetic acid and lower growth on glycerol medium (Figure 1, Panel G), phenotypes again restored by the expression of the full-length gene *ScLSM4*. Altogether, these results show that the *Sclsm4Δ1* mutant recapitulates all the phenotypes shown by *Kllsm4Δ1*, with regard to regulated cell death and premature aging [32], and also shows high sensitivity to nitrogen starvation that could be linked to a defect in the autophagy process. These phenotypes are not due to a lower expression of the *Sclsm4*Δ*1* gene, which, on the contrary, is highly expressed (Appendix A). In all the following experiments, as a wild-type strain, we used the *S. cerevisiae* CML39-11A strain. This strain is isogenic to MCY4 strains, except for the presence of *LSM4* under the Gal1-10 promoter, and we showed that it behaves as the MCY4 strain expressing the *LSM4* full-length gene [8,33,34].

### 2.2. The Sclsm4Δ1 Mutant Is Defective in Autophagy

We have previously reported that the majority of the mutant phenotypes in the *S. cerevisiae LSM4* mutant expressing *Kllsm4*Δ*1* can be suppressed by over-expressing *NEM1*, which codes for the catalytic subunit of the yeast nuclear membrane-resident protein phosphatase complex Nem1/Spo7 [35]. Nem1 is said to be necessary for the induction of autophagy after TORC1 inactivation [36], so we used a GFP-Atg8 processing assay to determine whether *LSM4* was involved in the macroautophagy flux [37].

The strains of interest (wild type and *Sclsm4Δ1* mutant) were transformed with pUG36/ATG8 in order to synthesize the chimerical protein GFP-Atg8. As the GFP β-barrel structure is more resistant than Atg8 to vacuolar hydrolysis, the presence of free GFP on the Western blot indicates that the autophagic process has occurred. Autophagy can be induced under a variety of deprivation conditions, such as the depletion of nitrogen and post-diauxic shift [38]. The autophagic flux has been evaluated upon nitrogen deprivation and during the post-diauxic growth phase, both in wild-type and *Sclsm4Δ1* mutant cells expressing the fusion protein GFP-Atg8. To exclude the possibility that a misregulation in the autophagy pathway could lead to an improper activation of the *MET17* on the pUG36/ATG8 plasmid, affecting the GFP-Atg8 processing assay itself, cells were grown on SD medium supplemented with auxotrophic requirements with the addition of methionine.

As shown in Figure 2A, during exponential growth, autophagy was not observed in the wild type nor in the mutant (Exp, lanes 1 and 4), as only the GFP-Atg8 fusion protein was detected. Protein extracts obtained from cells in the post-diauxic phase (PD, lanes 2 and 5) and nitrogen starvation (SD-N, lanes 3 and 6) showed that free GFP production was reduced in *Sclsm4Δ1* cells, suggesting a defect in this mutant in inducing macroautophagy. Interestingly, the fusion protein GFP-Atg8 accumulates at a higher level in the mutant both in the exponential and post-diauxic phases, suggesting a stabilization at the mRNA or protein level that needs to be investigated in the future.

Macroautophagy is important for survival during nutrient starvation, and mutants defective in autophagy rapidly lose cell viability after nitrogen starvation [39]. In fact, the majority of the phenotypes linked to autophagy mutants may actually be explained, at least in part, by the defective autophagy cells’ inability to synthesize new proteins and their failure to maintain physiological levels of amino acids [40].

From the GFP-Atg8 assay, the *Sclsm4Δ1* mutant seemed to have important defects in inducing autophagy, and the short CLS in the nitrogen starvation condition previously demonstrated (Figure 1F) supports this hypothesis.

### 2.3. Low Doses of Rapamycin Cannot Protect Cells from Oxidative Stress

As previously demonstrated for *Kllsm4Δ1* [8], the MCY4/*Sclsm4Δ1* strain shows high sensitivity to oxidative stress, and the viability is restored by the expression of the full-length gene *ScLSM4* (Figure 3A). Oxidative stress can induce autophagy both in yeast and in mammalian cells [41]. At the same time, it has been reported that rapamycin-induced autophagy confers neuroprotection against aging-induced oxidative stress in old rats [42]. Therefore, we first evaluated the cells’ response to the antibiotic rapamycin. We found that the *Sclsm4Δ1* mutant was highly sensitive even to low doses of rapamycin, in that exponentially growing cells exposed to 6 nM rapamycin reduced cell viability to 4% within 4 h, while this was not the case for the wild-type strain, which maintained cell viability equal to untreated cells (Figure 3B).

Similar sensitivity to this drug was found in the *lsm1*Δ mutant, which is a component of the heptameric ring-shaped complex formed by Lsm1 to Lsm7 [43]. As reported in Appendix A, the serial dilution assay showed the high sensitivity to 6 nM rapamycin of both *Sclsm4Δ1* and *lsm1Δ* cells compared to their respective wild types, CML39-11A and BMA38, suggesting that in both LSM mutant strains, autophagy is impaired.

Then, we explored if low doses of rapamycin could protect cells from hydrogen peroxide-induced cell death. As shown in Figure 3C, the presence of 6 nM rapamycin did not protect wild-type cells from oxidative stress, as the differences in viability of the treated and untreated samples after exposure to different concentrations of H_2_O_2_ were not statistically significant. On the other hand, it was not possible to evaluate the protective action of rapamycin in the *Sclsm4Δ1* mutant due to its high toxic effect. Similar results were obtained using lower doses of H_2_O_2_ as described in [44], confirming that the low dose of rapamycin used in this work is not sufficient to protect the aging-induced oxidative stress in the tested strains (Appendix A).

### 2.4. Sclsm4Δ1 Mutant Accumulated Cytoplasmic Autophagy-Related Structures

During autophagy, bulk cytoplasmic material is sequestered by the phagophore, a double-membrane structure that expands around the cargo, forming a sealed, double-membrane vesicle known as the autophagosome (AP). The autophagosome fusion to the vacuole leads to degradation and recycling of the cargo. Autophagic flux can be monitored by the localization of GFP-Atg8, which is delivered to the vacuoles to be degraded. With the aim of gaining more information on the autophagic step blocked in the *Sclsm4Δ1* mutant, we followed the localization of the fusion protein GFP-Atg8 by fluorescence microscopy.

During the exponential phase of growth in SD, around 1% and 7% of the WT and *Sclsm4Δ1* mutant cells, respectively, had a single GFP-Atg8 dot denoting the autophagosome localized near the vacuole membrane (Figure 4A, SD exp), with most of the fluorescence uniformly distributed into the cytoplasm. During this growth phase, in the *Sclsm4Δ1* mutant, a small percentage of cells show two or more GFP-Atg8 dots per cell (Figure 4A).

After 4 h of nitrogen starvation (SD-N), the differences between the wildtype and the mutant increased, with a mean percentage of GFP-Atg8 dots around 3% for the wild type and 25% for the *Sclsm4Δ1* mutant (Figure 4A, SD-N 4h). Moreover, the number of cells showing more than or equal to two GFP-Atg8 dots increased in the mutant cells to about 12%, representing half of the cell population with GFP-Atg8 dots (Figure 4A).

In the post-diauxic phase, there was a little increase in autophagy (Figure 2A), and, as expected, most of the GFP was localized inside the vacuole in the wild type (Figure 4C, PD), with about 3% of cells showing GFP-Atg8 dots. Concerning the *Sclsm4Δ1* mutant, the percentage of cells showing GFP-Atg8 dots increased up to about 20%, with very few cells showing intravacuolar fluorescence. These results reflect the lower autophagy observed in Figure 2B.

After 16 h of nitrogen starvation (Figure 4C, SD-N 16 h), the GFP is predominantly found in the vacuole in the wild-type cells, confirming the active autophagic flux, with only a slight increase in cells showing cytoplasmic dots, while about 35% of *Sclsm4Δ1* mutant cells showed cytoplasmic dots, and half of them presented two or more dots per cell (Figure 4D) and very low intravacuolar fluorescence.

After three days of growth in SD, more than 90% of *Sclsm4Δ1* mutant cells showed GFP-Atg8 dots, representing the number of wild-type cells presenting dots in about 13% of the population (Figure 4E, quantification in 4F). This percentage increased a little bit in the wild type incubated for 3 days in SD-N medium, while in the *Sclsm4Δ1* mutant cells, those presenting GFP-Atg8 dots after 3 days in SD-N medium were the same as after 16 h of incubation in SD-N (about 40%, Figure 4F). This could be due to the rapid loss of viability of the *Sclsm4Δ1* mutant in SD-N observed already at day 1 (Figure 2C). These data, altogether, indicate that *Sclsm4Δ1* mutant cells accumulated autophagy-related structures when autophagy was induced by nitrogen starvation or during aging, and further vacuolar co-localization analysis is planned to confirm this evidence.

Similar outcomes were obtained from the same experiments using the *lsm1*Δ mutant in a different genetic background (Appendix A), indicating that the observed autophagy defects are a trait of LSM mutants.

Interestingly, the *lsm1Δ* mutant showed an overall lower percentage of cells with GFP-Atg8 dots compared to the *Sclsm4Δ1* mutant, in particular in the stationary phase in both the SD and SD-N media, but with a higher component of two or more dots per cell. Furthermore, the correspondent wild type (BMA38) showed a slightly lower percentage of cells with GFP-Atg8 dots compared to the wild type CML39-11A, especially in SD-N after 16 h of incubation, suggesting a difference in autophagosome formation or transport due to the genetic background.

## 3. Discussion

The *Sclsm4Δ1* mutant of *S. cerevisiae*, which expresses a truncated form of the essential gene *LSM4*, showed premature aging, fragmented nuclei, and ROS accumulation. The *Sclsm4Δ1* mutant also showed high sensitivity to the regulated cell death-inducers acetic acid and caffeine, the same phenotypes as those previously shown for the *Kllsm4Δ1* mutant of *K. lactis* [6,7,8,45]. It has been reported that mutants hypersensitive to caffeine can also have defects in autophagy [46].

Human *LSM4* was indicated among the Differentially Expressed Autophagy-Related Genes (DE-ARGs) in a study that aimed to find interactions between autophagy and hepatocellular carcinoma (HCC) pathogenesis [47], but, up to now, there has been no evidence of the direct involvement of *LSM4* in autophagy [47]. Our results shown here demonstrate, by means of the GFP-Atg8 fusion protein, that a defect in autophagy induction upon nitrogen starvation was present in the *Sclsm4Δ1* mutant.

The *Sclsm4*Δ*1* mutant’s rapid viability loss in the presence of low doses of rapamycin and nitrogen starvation is evidence that autophagy induction is defective in this mutation.

Rapamycin is known to have a protective effect against oxidative stress, but at a 6 nM concentration, it did not exert a protective effect on the wild-type cells challenged with hydrogen peroxide, probably because the rapamycin concentration employed in this experiment was too low. In the case of the *Sclsm4Δ1* mutant, it was not possible to draw any conclusions due to the elevated toxicity of rapamycin in this mutant. High rapamycin sensitivity was also observed in the deletion mutant of *LSM1*, encoding a unit of the heptameric ring-shaped complex formed by Lsm1 to Lsm7, which, together with Pat1, is involved in mRNA degradation.

We also followed the GFP-Atg8 fusion protein, a marker for phagophore assembly site (PAS) and autophagosome formation, using fluorescence microscopy to look into the autophagic flux in our mutants. We showed that GFP-Atg8 dots accumulate in the *Sclsm4Δ1* mutant during nitrogen starvation, the diauxic phase, and aging.

It has been reported that under starvation conditions, cell death in autophagy-defective yeast mutants is caused by mitochondrial dysfunction [48]. Actually, mitochondrial defects were described for the *Kllsm4Δ1* mutant in that it accumulated ROS, showed growth arrest on respiratory carbon sources, and had an aberrant mitochondrial morphology with a punctuate distribution instead of the normal tubular shape [8,14]. Interestingly, the defects in mitochondrial morphology and the subsequent growth arrest on glycerol are reduced by the overexpression of *HIR1*, encoding for the co-repressor of histone gene transcription, *PGK1*, encoding the glycolytic enzyme phosphoglycerate kinase 1, *NEM1*, the catalytic subunit of the Nem1/Spo7-Pah1 axis, and by acetyl-L-carnitine supplementation [33,34,35,49]. Moreover, from genome-wide studies, it has also been found that the *lsm1Δ* mutant cannot grow on glycerol [50], suggesting a probable link between mRNA metabolism and respiration, but the mechanisms by which the decapping mutants arrest to grow on respiratory carbon sources are still under research.

Given that oxidation has been shown to inhibit the Atg4 protease activity in a H_2_O_2_ concentration-dependent manner, *Sclsm4Δ1* may accumulate ROS intracellularly and exhibit defects in autophagy induction [51]. Nevertheless, autophagosome formation is abolished in *atg4Δ* cells [52], while the *Sclsm4Δ1* mutant, as well as the *lsm1Δ* mutant, showed an accumulation of GFP-Atg8 dots, suggesting that the autophagic defects are principally due to defects in a late stage of autophagosome formation preceding the fusion of mature autophagosomes with the vacuole or in the autophagosome-vacuole fusion process itself.

There is evidence that the Pat1-Lsm complex could be involved in these steps, as, upon nitrogen starvation, the Pat1-Lsm complex binds and stabilizes a subset of *ATG* mRNA by preventing their exosome-mediated 3’→5’ degradation. Among these is Atg1, a serine/threonine kinase homolog to human ULK-Kinase [28,41] that is considered a key regulator of autophagy. Atg1 phosphorylates the Atg4 protease, keeping it inactive and preventing the premature release of Atg8 from autophagic membranes, and Ykt6 keeps this SNARE in an inactive state and so regulates the autophagosome-vacuole fusion [53,54]. Another Atg1 target is Vps34, a class III phosphatidylinositol 3-kinase whose phosphorylation is important for full autophagy activation and cell survival [55]. Vps34 was mislocalized in mutants of the Nem1/Spo7-Pah1 axis but localized at the right compartments after rapamycin treatment, suggesting that the Nem1/Spo7 complex supports autophagy induction after TORC1 inactivation by nutrient starvation, probably via membrane synthesis [36].

We previously reported that in the *Kllsm4Δ1* mutant, the ER appears aberrant, and the overexpression of *NEM1*, the catalytic subunit of the Nem1/Spo7-Pah1 axis, could rescue the *Kllsm4Δ1* mutant phenotypes, suggesting that the Nem1/Spo7-Pah1 axis could be compromised in the *Sclsm4Δ1* mutant [35].

The defects in the Nem1/Spo7-Pah1 axis, together with the possible high degradation of *ATG1* mRNA, could concur with the observed autophagy defects in the *Sclsm4Δ1* mutant. Nevertheless, to date, it is still not possible to determine which pathway is affected by the observed *Sclsm4Δ1* mutant defects, and further investigations will be needed to clarify this important point.

It has been recently reported that the phosphorylation of Edc3, a P-body component, has an effect on tumor growth and invasion through controlling P-bodies formation and dynamics [56]. In a genome-wide analysis, it has been reported that Atg1 could also phosphorylate Lsm4 [57]. As *LSM4* is also involved in some cancers, it will be interesting to use the simple yeast model to find *LSM4* targets for the development of antitumoral molecules.

## 4. Materials and Methods

### 4.1. Yeast Strains, Growth Conditions, and Plasmid Construction

*S. cerevisiae* strains used in this work are described in Table 1. Cells were grown at 28 °C in YPD (1% yeast extract Gibco, Thermo Fisher Scientific, Inc., Waltham, MA, USA, #212750), 2% bacto-peptone Gibco, Thermo Fisher Scientific, Inc., Waltham, MA, USA, #211677 ), 2% glucose) and SD (0.67% yeast nitrogen base without aminoacids (Becton, Dickinson and Company, 1 Becton Drive Franklin Lakes, NJ, USA, #291940), 2% glucose) supplemented with auxotrophic requirements. YPY medium (1% yeast extract (Gibco, Thermo Fisher Scientific, Inc., Waltham, MA, USA, #212750, 2% bacto-peptone Gibco, Thermo Fisher Scientific, Inc., Waltham, MA, USA, #211677, 2% glycerol) was used to evaluate growth on glycerol as a carbon source. For autophagic induction by nitrogen starvation, cells were grown in SD-N (0.17% yeast nitrogen base without aminoacids and ammonium sulphate Becton, Dickinson and Company, 1 Becton Drive Franklin Lakes, NJ, USA, #233520, 2% glucose). Solid media were obtained by the addition of 2% Bactoagar Becton, Dickinson and Company, 1 Becton Drive Franklin Lakes, NJ, USA, #214010.

The plasmid pRS313/*Sclsm4Δ1* was obtained by amplifying 868 bp of the *ScLSM4* gene, comprising the promoter region and the gene portion encoding the first 84 aminoacids, and then cloning the PCR fragment with BamH1/SacI extremities in the specific site of the vector (primers listed in Table 2). *E. coli* DH5α cells were used to amplify the vector. MCY4 transformation to give the strain MCY4/*Sclsm4Δ1* was performed by the ONE-STEP method [59] with ONE-STEP buffer (PEG 3350 40%, LiAc 0.2 M, DTT 0.1 M, and ssDNA carrier 0.1 μg/μL (Sigma-Aldrich, Darmstadt, Germany, D1626)) as the transformation mix.

The plasmid pRS313/*ScLSM4* was obtained by amplifying 1308 bp of the *ScLSM4* gene, comprising the promoter region and the complete coding region, and then cloning the PCR fragment with BamH1/SacI extremities in the specific site of the vector (primers listed in Table 2). *E. coli* DH5α cells were used to amplify the vector. MCY4 transformation was performed by the ONE-STEP method [59] to give the strain MCY4/*ScLSM4.*

Plasmid pUG36/*ATG8* was provided courtesy of Tobias Eisenberg and colleagues [60]. Transformation of the selected strains was performed by the ONE-STEP method [59].

To determine the growth rates in rich medium, MCY4/*Sclsm4Δ1*, MCY4/*ScLSM4*, and CML39-11A strains were grown exponentially on YPD, and OD_600_ values were taken every two hours. The growth rate (μ) was calculated as (ln Nt − ln N_0_) / (t − t_0_). Results are reported in Appendix A.

### 4.2. Viability Assays

Stationary cultures of strains MCY4/*Sclsm4Δ1* and CML39-11A were tested for microcolony-forming ability during the chronological lifespan in SD and SD-N media. Starting from an overnight preculture in 3 mL of SD with auxotrophic requirements, cells were counted and diluted to a final concentration of 5 × 10^5^ cells/mL in 20 mL of medium (SD or SD-N), and the flasks were incubated at 28 °C. Starting from day 1, 3 × 10^4^ cells were daily plated on a YPD-coated slide and analyzed with an optic microscope after 1–2 days of incubation at 28 °C. Cell viability was calculated as the percentage of microcolony-forming cells [61].

### 4.3. Rapamycin Treatment

Strains MCY4/*Sclsm4Δ1* and CML39-11A were tested for the microcolony-forming ability after treatment with 6 nM of rapamycin (Sigma-Aldrich, Darmstadt, Germany, R8781). Starting from an exponential preculture in 40 mL of SD medium with auxotrophic requirements (0.2–0.3 OD_600_), the cells were split in 20 mL SD medium with auxotrophic requirements (controls) or SD with auxotrophic requirements supplemented with 6 nM of rapamycin (test samples). After 1, 2, 3 and 4 h of treatment, 3 × 10^4^ cells were plated on a YPD-coated slide and analyzed with an optic microscope after 1–2 days of incubation at 28 °C. Cell viability was calculated as the percentage of microcolony-forming cells.

### 4.4. Fluorescence Microscopy

Nuclear morphology was detected with the DAPI staining (1 μg/mL) (Sigma-Aldrich, Darmstadt, Germany, D8417) of 1 mL of exponentially growing cells (0.2–0.4 OD_600_) fixed with 70% (*v*/*v*) ethanol. Oxygen reactive species (ROS) were detected by incubating 1 mL of cells with 5 μg/mL of DHR (Sigma-Aldrich, Darmstadt, Germany, D1054) for 4 h at 28 °C and then analyzed using fluorescence microscopy (Axioskop 2, Carl Zeiss, Germany). The vacuolar membranes were detected by incubating 1 mL of cells with FM4-64 (Invitrogen, Thermo Fisher Scientific, Inc., Waltham, MA, USA, T13320) at a final concentration of 2 μM for 10 min at 28 °C. The visualization of autophagosome formation and translocation was performed using the reporter plasmid pUG36/*ATG8* and analyzed using fluorescence microscopy. The strains of interest were grown in 20 mL of SD medium supplemented with auxotrophic requirements and methionine (10 μg/mL), then harvested at their exponential phase (0.2–0.4 OD_600_), washed with H_2_O, and split in 10 mL of SD medium supplemented with auxotrophic requirements and methionine (10 μg/mL) and 10 mL of SD-N medium without aminoacids. Flasks were incubated at 28 °C and then harvested at their logarithmic growth phase (Exp, 0.4 OD_600_), post-diauxic phase (after 16 h, 0.9–1 OD_600_), and stationary phase (after 3 days, 1.5 OD_600_). The percentage of GFP-Atg8 dots-positive cells was calculated among the total number of fluorescent-positive cells. The cells were counted manually with ImageJ software (Version 1.8.0_172) [62].

### 4.5. H_2_O_2_ Sensitivity Test

Starting from an exponential preculture in 20 mL of SD medium with auxotrophic requirements (0.2–0.3 OD_600_), 1 mL of culture was incubated for 4 h at 28 °C with 0 mM (control), 0.8 mM, 1.2 mM, and 3 mM of H_2_O_2_ (Sigma-Aldrich, Darmstadt, Germany, #216763), and then 3x10^4^ cells were plated on a YPD-coated slide and analyzed with an optic microscope after 24 h of incubation at 28 °C. Cell viability was calculated as the percentage of microcolony-forming cells. To check cellular response to lower concentrations, the same experiments were performed, exposing cells to H_2_O_2_ at concentrations of 0.1 mM, 0.2 mM and 0.5 mM, as described in [44]. The results are presented in Appendix A.

### 4.6. Glycerol Growth, Caffeine, Acetic Acid, and Rapamycin Sensitivity Test

Serial dilutions of strains CML39-11A and MCY4/*Sclsm4Δ1* were spotted on YPD, YPY, YPD + 0.25% caffeine, YPD + 60 mM acetic acid, and YPD + 6 nM rapamycin, and their viability was detected after 2–3 days of incubation at 28 °C. Serial dilutions of strains BMA38 and *lsm1*Δ were spotted on YPD and YPD + 6 nM rapamycin, and their viability was detected after 2–3 days of incubation at 28 °C.

### 4.7. Protein Extraction and Western Blot Analysis of Autophagy-Induced Cells

The decapping mutant and wild-type strain were grown on 20 mL of SD medium supplemented with auxotrophic requirements and methionine (10 μg/mL), then harvested at their exponential phase (0.2–0.4 OD_600_), washed with H_2_O, and split in 10 mL of SD medium supplemented with auxotrophic requirements and methionine (10 μg/mL) and 10 mL of SD-N medium without aminoacids. Flasks were incubated at 28 °C and then harvested at their logarithmic growth phase (Exp, 0.4 OD_600_) and post-diauxic phase (after 16 h, 0.9–1 OD_600_). The amount of cells corresponding to 2 OD_600_ was washed with H_2_O, resuspended in 200 μL of NaOH 2 M/β-mercaptoethanol 5% and then chilled on ice for 10′. Protein precipitation was performed with TCA at a final concentration of 8.3%, centrifugation at 13,000 rpm for 15′ and pellet-suspended in 100 μL of loading buffer (50 mM Tris-HCl pH 6.8; 100 mM β-mercaptoethanol; 2% SDS, 0.1% bromophenol blue; 10% glycerol). Samples were then boiled at 95 °C for 5′ and loaded into a 12% acrylamide SDS-PAGE gel. A protein marker was loaded in the first lane (Thermo-Fisher, Thermo Fisher Scientific, Inc., Waltham, MA, USA, LC5925). Separated proteins were transferred onto nitrocellulose membrane through electroblotting. Ponceau red staining was used as a loading control (0.1% Ponceau S (Sigma-Aldrich, Darmstadt, Germany, P-3504), 5% acetic acid). Autophagic cargo processing was studied via immunoblotting analysis using an anti-GFP antibody (α-mouse-GFP, Santa Cruz Biotechnology, Santa Cruz Biotechnology, Dallas, TX, USA, sc-9996) to detect GFP-Atg8, as described [35]. The secondary HRP-associated antibody was sc-2060 Santa Cruz anti-mouse (goat) (Santa Cruz Biotechnology, Dallas, TX, USA). The percentage of autophagy activation was determined as the ratio between free GFP and the total GFP (free GFP + fusion protein GFP-Atg8), calculated with the Image Lab^TM^ Volume Tool Software (Version 6.1.0, Bio-Rad, Hercules, CA, USA) after image capture at the ChemiDocTM XRS+ System (Bio-Rad, Hercules, CA, USA).

### 4.8. RNA Extraction, cDNA Syntesis, and Real-Time qPCR for mRNA Expression of ScLSM4

Strains MCY4/*Sclsm4Δ1*, MCY4/*ScLSM4*, and CML39-11A were grown on 20 mL of YPD and harvested at their exponential phase (0.2–0.4 OD_600_). An amount of cells corresponding to 4 OD_600_ was washed with H_2_O, resuspended in 200 μL of lysis buffer (0.5 M NaCl, 0.2 M Tris-HCl pH 7.5, 10 mM EDTA, 1% SDS) and 200 μL of phenol-chlorophorm-isoamyl alcohol (PCI) 25:24:1 (Sigma-Aldrich, Darmstadt, Germany, 77617) and grounded by vortexing with micro glass beads. After the addition of 300 μL of lysis buffer and 300 μL of PCI, cells were centrifuged at 10,000 rpm for 5′ at 4 °C, and then the supernatant was precipitated with 3 volumes of EtOH at −20 °C for 30′. The precipitated nucleic acids were resuspended in 15 μL of RNase-free H_2_O. The integrity of RNA was tested via electrophoresis on agarose gel, 1% in TAE buffer 1× (40 mM Tris, 20 mM Acetate, and 1mM EDTA). RNA was treated with DNaseI using the DNA-*free*
^TM^ kit (Invitrogen, Thermo Fisher Scientific, Inc., Waltham, MA, USA AM1906) and retrotrascribed in cDNA using the SensiFAST cDNA Synthesis Kit (Meridian Bioscience, Inc., Cincinnati, OH, USA, BIO-65053), according to their datasheets, respectively. To evaluate the expression levels of *LSM4*, the obtained cDNAs were used as a template for a Real-Time qPCR assay using the primers listed in Table 3, using the SensiFAST SYBR Hi-ROX kit (Meridian, Bioscience, Inc., Cincinnati, OH, USA, BIO-92020). The *TDH3* gene was used as the calibrator. Data were obtained using StepOne Plus (Applied Biosystem, Thermo Fisher Scientific, Inc., Waltham, MA, USA) and further analyzed with the ΔΔCt method. The results are reported in Appendix A.

### 4.9. Statistical Analysis

Figure 1E,F, Figure 2, Figure 3 and Appendix A show the mean of three independent experiments. The bar error indicates standard deviation. Figure 1A–D, Figure 4 and Appendix A show the mean of three independent experiments, with 700 cells per set (Figure 1A–D) and >300 cells per set (Figure 4 and Appendix A). Appendix A shows the mean of two independent experiments. The bar error indicates the standard deviation. To evaluate the statistical significance, a two-tailed, two-sample unequal variance test was performed, and the number of stars (*) indicates the p-value range. * *p*-value < 0.05, ** *p*-value < 0.01, *** *p*-value < 0.001, **** *p*-value < 0.0001, no star: no statistically significant.

## 5. Conclusions

As a useful model system for aging and aging-associated pathologies, our research group has been studying a specific yeast mutant that shows premature aging [6]. This *Saccharomyces cerevisiae* strain expresses a truncated form of the essential protein Lsm4 of *Kluyveromyces lactis* (*Kllsm4Δ1*), while the expression of the endogenous *LSM4* is repressed. This protein is a component of the *LSM* complexes that are essential for the splicing process in the nucleus and RNA degradation in the cytoplasm, and the expression of the truncated form restores the viability of the strain while leading to premature markers of aging, such as ROS accumulation and nuclei fragmentation, and regulated cell death [6,7]. In addition, RNAs accumulate in the cytoplasm due to the lack of the C-terminal Q/N-rich domain of *KlLSM4* that is needed for efficient RNA degradation [7,8] and for P-bodies localization [9,10]. The expression of the truncated *S. cerevisiae* protein in the absence of *LSM4* recapitulated all the phenotypes observed in the *Kllsm4Δ1* mutant [6], suggesting a role for the *LSM4* C-terminus in maintaining viability during CLS. We found that the *Sclsm4Δ1* mutant showed defects in the induction of autophagy and was very sensitive to nitrogen starvation or treatment with low doses of rapamycin. This could be explained by a misregulation of the Autophagy-Related Genes (ATG) mRNAs together with the defect in the Nem1/Spo7-Pah1 axis that we previously demonstrated for the *Kllsm4Δ1* mutant, and further experiments are needed to confirm the involvement of Lsm4 in this process.

Moreover, both during nitrogen starvation and aging, the *Sclsm4Δ1* mutant and another mutant of the cytosolic LSM complex in a different genetic background, *lsm1*Δ, accumulated cytoplasmic autophagy-related structures, suggesting a role for Lsm4 and the LSM complex in later stages of autophagosomes internalization.

## Figures and Tables

**Figure 1 ijms-24-13708-f001:**
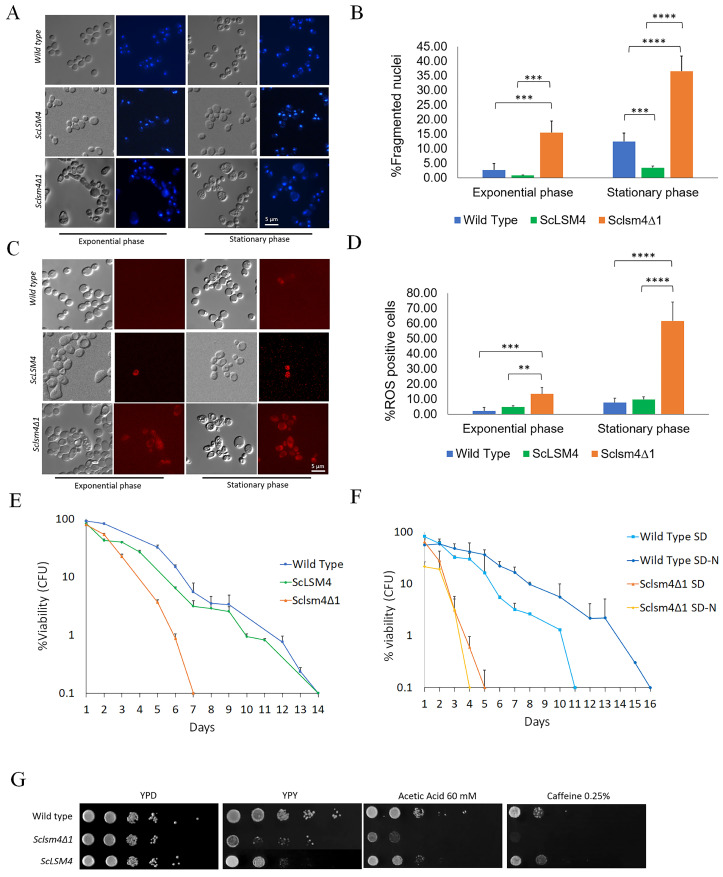
The *Sclsm4Δ1* mutant recapitulates the same pleiotropic phenotypes as described for the *Kllsm4Δ1* mutant. (**A**) DAPI staining of the CML39-11A (wild type), MCY4/*Sclsm4Δ1* mutant cells, and MCY4/*ScLSM4* in both exponential and stationary phases. (**B**) Percentage of fragmented nuclei over total cells for the same strains and conditions as in (**A**) from three independent experiments. (**C**) Dihydrorhodamine 123 (DHR123) staining of the CML39-11A (wild type) and MCY4/*Sclsm4Δ1* mutant cells and MCY4/*ScLSM4* in both exponential and stationary phases. (**D**) Percentage of ROS-positive cells over total cells for the same strains and conditions as in (**C**) from three independent experiments. Data are represented as the mean percentage of 700 cells per set ± standard deviation. (**E**) Chronological life span of the CML39-11A- (wild type) and MCY4-expressing *Sclsm4Δ1* mutant or the full-length *LSM4* protein (*ScLSM4*) cells. Data are represented as the mean of three independent experiments ± standard deviation. (**F**) Chronological life span of CML39-11A (WT) and mutant *Sclsm4Δ1* cells cultured in standard synthetic medium (SD) or in nitrogen-deprived medium (SD-N). Data are represented as the mean of three independent experiments ± standard deviation. (**G**) 10-fold dilutions were spotted on complete solid media containing 2% glycerol (YPY), YPD containing 60 mM acetic acid, and 0.25% caffeine, and plates were incubated at 28 °C for 3 days. YPD was used as a growth control. ** *p*-value < 0.01, *** *p*-value < 0.001, **** *p*-value < 0.0001.

**Figure 2 ijms-24-13708-f002:**
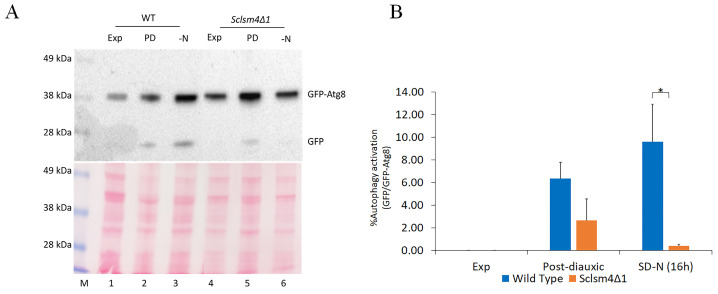
The *Sclsm4Δ1* mutant shows defects in autophagy induction. (**A**) CML39-11A (WT) and mutant *Sclsm4Δ1* cells were grown exponentially in SD medium (Exp), then the same amount of cells was centrifuged and resuspended in SD and in SD-N (nitrogen deprivation, -N) medium and further incubated for 16 h (PD: post-diauxic phase). Ponceau red staining has been used as a load control. One of three independent experiments is shown. (**B**) The percentage of autophagy activation was measured as the ratio between free GFP and GFP-ATG8 in three independent experiments. * *p*-value < 0.05. Data are represented as the average and standard deviation of three independent experiments, and one representative experiment is shown. * *p*-value < 0.05.

**Figure 3 ijms-24-13708-f003:**
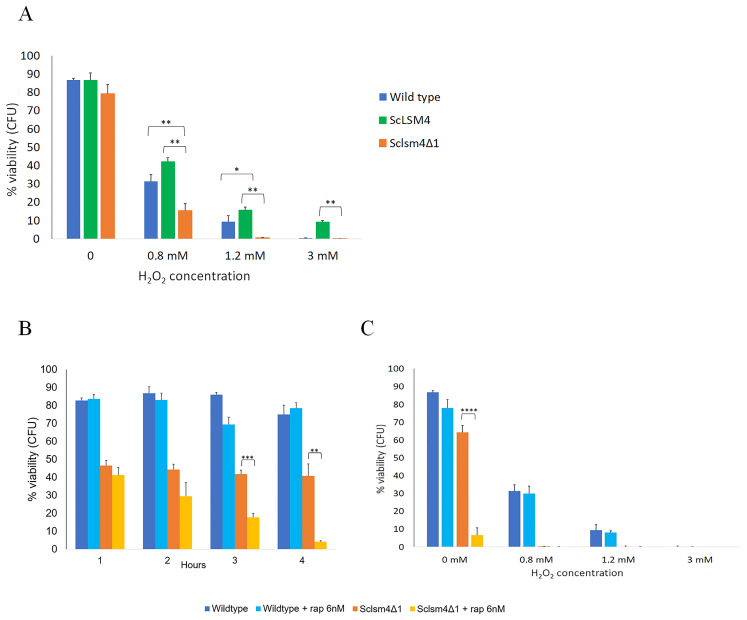
Treatment with a low dose of rapamycin does not protect the cells from oxidative stress. (**A**) Cell viability of the CML39-11A (wild type) and MCY4-expressing *Sclsm4Δ1* mutant or the full-length Lsm4 protein (*ScLSM4*) cells was measured after exposure to H_2_O_2_ at the indicated concentrations for 4 h. (**B**) Cell viability of the CML39-11A (wild type) and MCY4-expressing *Sclsm4Δ1* was measured every hour after treatment with 6 nM of rapamycin in SD medium for 4 h. (**C**) Cell viability of the CML39-11A (wild type) and *Sclsm4Δ1* mutants was measured after exposure to H_2_O_2_ at the indicated concentrations for 4 h. In total, 6 nM rapamycin was added 4 h prior to exposure to H_2_O_2_. Data are represented as the mean of three independent experiments ± standard deviation. * *p*-value < 0.05 ** *p*-value < 0.01 *** *p*-value < 0.001 **** *p*-value < 0.0001.

**Figure 4 ijms-24-13708-f004:**
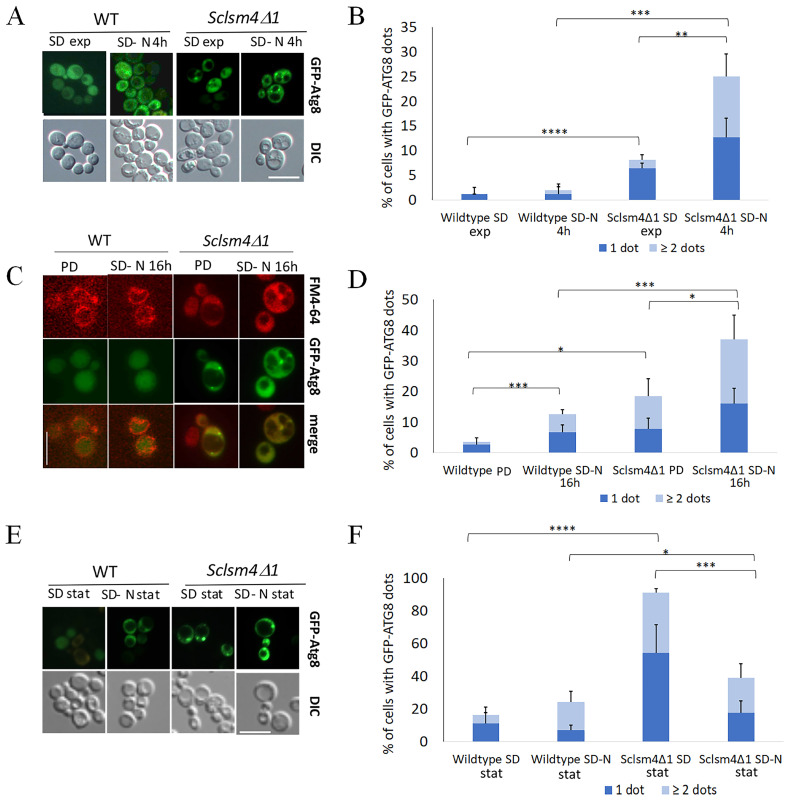
The *Sclsm4Δ1* mutant shows defects in autophagy-related structure transport to the vacuole, as indicated by a higher percentage of GFP-ATG8 dots in the cytoplasm during nitrogen starvation and CLS. Wild-type CML39-11A and mutant *Sclsm4Δ1* cells expressing the fusion protein GFP-ATG8 were observed at the fluorescence microscope during the exponential phase in both SD and SD-N medium for 4 h (**A**), during the post-diauxic phase (PD), in SD-N for 16 h (**C**) and after 3 days of growth in SD (SD stat) or SD-N (SD-N stat) (**E**). GFP-Atg8 dots per cell were quantified from three biological replicates (n ≥ 300 cells), and the mean of cells containing one or more than or equal to two dots is plotted in (**B**,**D**,**F**). Error bars represent the standard deviation. * *p*-value < 0.05 ** *p*-value < 0.01 *** *p*-value < 0.001 **** *p*-value < 0.0001. Scale bar: 5 μm.

**Table 1 ijms-24-13708-t001:** *S. cerevisiae* strains used in this work.

Strain	Genotype	Source
MCY4	*MATα*, *ade1-101, his3-*Δ*1*, *trp1-289*, *ura3*, *LEU-GAL1-SDB23*	[58]
MCY4/*Sclsm4Δ1*	*MATα*, *ade1-101*, *his3-*Δ*1*, *trp1-289*, *ura3, LEU-GAL1-SDB23* pRS313/*Sclsm4Δ1*	This work
CML39-11A	*MAT*a, *ade1-101, his3-*Δ*1, leu2*, *ura3, trp1-289*	[8]
MCY4/*ScLSM4*	*MATα*, *ade1-101*, *his3-*Δ*1*, *trp1-289*, *ura3*, *LEU-GAL1-SDB23* pRS313/*ScLSM4*	This work
MCY4/*Sclsm4Δ1* pUG36/*ATG8*	*MATα, ade1-101*, *his3-*Δ*1*, *trp1-289*, *ura3*, *LEU-GAL1-SDB23* pRS313/*Sclsm4Δ1*, pUG36/*ATG8*	This work
CML39-11A pUG36/*ATG8*	*MATa*, *ade1-101*, *his3-*Δ*1*, *leu2*, *ura3*, *trp1-289* pUG36/*ATG8*	This work
MCY4/Sc*LSM4* pUG36/*ATG8*	*MATα*, *ade1-101*, *his3-*Δ*1*, *trp1-289*, *ura3*, *LEU-GAL1-SDB23* pRS313/*ScLSM4*, pUG36/*ATG8*	This work
BMA38	*MATα*, *ura3-1*, *leu2-3*, *-112*, *ade2-1*, *can1-100*, *his3-11*, *-15, trp1*Δ*1*	[39]
BMA38 lsm1Δ	*MATα*, *ura3-1*, *leu2-3*, *-112*, *ade2-1*, *can1-100*, *his3-11*, *-15*, *trp1*Δ*1*, *lsm1*Δ*::TRP1*	[39]

**Table 2 ijms-24-13708-t002:** Amplification and cloning of the N-terminus truncated *ScLSM4* gene.

Primer Name	Oligonucleotide Sequence
*BamH1-ScLSM4/Sclsm4Δ1* Fw	5’-AAAAAAGGATCCGTACGCAGTCACAATGCGG-3’
*SacI-ScLSM4* Rv	5’-GGGGGGAGCTCACCTGTAAACTAAAGGAAAGCTCG-3’
*SacI-Sclsm4Δ1* Rv	5’-GGGGGGAGCTCTTATCTTGCAATTTGATAAACTTGATAAAAGTCC-3’

**Table 3 ijms-24-13708-t003:** Real-time qPCR for *ScLSM4* gene.

Primer Name	Oligonucleotide Sequence
*ScLSM4 N-term Fw*	5’-ATTGACCAACGTAGATAACTGGA-3’
*ScLSM4 N-term Rv*	5’-TACGGCTTTACTGCTCTCAG-3’
*TDH3 Fw*	5’-CGGTAGATACGCTGGTGAAGTTTC-3’
*TDH3 Rv*	5′-TGGAAGATGGAGCAGTGATAACAAC-3′

## Data Availability

Data are contained within the article or Appendix A.

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
