# Peer review of "Yeast Lsm Pro-Apoptotic Mutants Show Defects in Autophagy"

_ijms, 2023, doi:10.3390/ijms241813708_

Round 1
Reviewer 1 Report
The authors Caraba et al., have submitted an original research manuscript entitled Yeast lsm pro-apoptotic mutants show defects in autophagy induction. In this study, the authors build on their previous results that demonstrated that Lsm4 protein is essential for cell viability. The authors go on to show results that demonstrate that the Lsm4 mutant proteins also have defective autophagy through different assays. While the study adds knowledge to the role of Lsm4 in regulating autophagy, multiple scientific and technical concerns remain. Some of the major comments on the manuscript are:
Apart from the assays of FIgure 1E, F, the MCY4 strain complemented with a full-length ScLSM4 should also be used for other assays of Figure 1. This will help to better appreciate the effect of mutation of the C-terminal domain of Lsm4.
The authors need to justify their use of a different strain with LSM4 mutation instead of using the MCY4 strain for the assay in Figure 3B.
For the assay in Figure 4, the authors could cite a study that has used similar concentrations of H2O2 with S. cerevisiae strains of similar background to rule out the possibility that the concentration of H2O2 is not detrimental to the viability of the cells. If so, the authors should test lower concentrations of H2O2 to study these effects while also confirming that the other hallmarks of H2O2 treatment occur at that lower concentration.
In Figure 5A, the presence of a GFP-Atg8 dot usually represents the formation of a mature autophagosome. Analysis of phagophore assembly site (PAS) localization of GFP-Atg8 would require high-resolution microscopy and needs to be confirmed by co-localization with other PAS marker proteins like Atg1. Also, the vacuolar proximity of the GFP-Atg8 cannot be commented on without the use of a vacuolar dye and co-localization analysis of GFP-Atg8 with this dye.
The representative Figure in Figure 5A doesn't reflect the statistics in Figure 5B. The authors should select more representative images.
The description of the results for the PD phase in Figure 5C and its correlation with Figure 2C should be elaborately explained including the details of the strains that are being referred to.
While some of the minor comments are:
The results of the Lsm4 RGG domain as explained in lines 51-54 are not contradictory. The cited articles and the text both indicate that Lsm4 RGG is involved but not essential for histone H2A degradation. The authors could revise this.
The authors have referred to the proteins with and without the letter "p". The authors should maintain consistency in their use of nomenclature throughout the manuscript.
The description of the results for Figure 1G could be reworded as this assay has only been tried with caffeine with K. lactis. Also, related studies could be cited here again.
Figures 1A and 1C need to improve in quality, especially for the channels depicting DAPI and DHR123 staining respectively.
Since Figures 2 and 3 both describe autophagy-related assays, they could be included in a single Figure.
The results on oxidative stress should be made a separate sub-section in the results.
The reference Figure for the description on lines 251-252 and 256-258 is Figure 5A and not 5B.
The Y-axis title of Figures 5B, 5D, and 5F should be renamed to "% of cells with GFP-Atg8 dots". Similarly, the description of the results should refer to the percentages as the number of cells with GFP-Atg8 dots
The authors could mention that the group in Figure 5B with -N is with 4h of treatment. Similarly, the group in 5D is with 16h of -N treatment. It would help to better appreciate their results across the entire Figure.
The English language requires moderate revision for a better understanding of the manuscript.
Author Response
We thank the reviewer for her/his constructive criticisms, which allowed us to improve our work. In the new manuscript, the changed parts are highlighted in yellow.
Below are the point-by-point answers to the reviewer’s criticisms.
Point 1.Apart from the assays of Figure 1E, F, the MCY4 strain complemented with a full-length ScLSM4 should also be used for other assays of Figure 1. This will help to better appreciate the effect of mutation of the C-terminal domain of Lsm4.
Response 1. We repeated the experiments also with the MCY4 strain complemented with a full-length ScLSM4 and inserted them in figure 1.
Point 2. The authors need to justify their use of a different strain with LSM4 mutation instead of using the MCY4 strain for the assay in Figure 3B.
Response 2. We used the wild type strain CML39-11A along all the experiments. This strain is isogenic to MCY4 strains, except for the presence of LSM4 under the Gal1-10 promoter and we showed that it behaves as the MCY4 strain expressing LSM4 full length gene in our previous works (Mazzoni C, Herker E, Palermo V, Jungwirth H, Eisenberg T, Madeo F, Falcone C. Yeast caspase 1 links messenger RNA stability to apoptosis in yeast. EMBO Rep. 2005 Nov;6(11):1076-81. doi: 10.1038/sj.embor.7400514. Epub 2005 Sep 9. PMID: 16170310; PMCID: PMC1371024; C. Mazzoni, V. Palermo, M. Torella, C. Falcone, HIR1, the co-repressor of histone gene transcription of Saccharomyces cerevisiae, acts as a multicopy suppressor of the apoptotic phenotypes of the LSM4 mRNA degradation mutant, FEMS Yeast Res. 5 (2005) 1229–1235. https://doi.org/10.1016/j.femsyr.2005.07.007 ; C. Mazzoni, M. Torella, A. Petrera, V. Palermo, C. Falcone, PGK1, the gene encoding the glycolitic enzyme phosphoglycerate kinase, acts as a multicopy suppressor of apoptotic phenotypes in S. cerevisiae, Yeast Chichester Engl. 26 (2009) 31–37. https://doi.org/10.1002/yea.1647). The lsm1Δ and its wild type strain BMA38 were a gift from J. Beggs laboratory (A.E. Mayes, L. Verdone, P. Legrain, J.D. Beggs, Characterization of Sm-like proteins in yeast and their association with U6 snRNA, EMBO J. 18 (1999) 4321–4331. https://doi.org/10.1093/emboj/18.15.4321).
Point 3. For the assay in Figure 4, the authors could cite a study that has used similar concentrations of H2O2 with S. cerevisiae strains of similar background to rule out the possibility that the concentration of H2O2 is not detrimental to the viability of the cells. If so, the authors should test lower concentrations of H2O2 to study these effects while also confirming that the other hallmarks of H2O2 treatment occur at that lower concentration.
Response 3. We used the same concentration of H2O2 with the same wild type strain (CML39-11A) presented in figure 4 in several studies, here we report the reference when it was used first, which we cited in this new version (Mazzoni C, Herker E, Palermo V, Jungwirth H, Eisenberg T, Madeo F, Falcone C. Yeast caspase 1 links messenger RNA stability to apoptosis in yeast. EMBO Rep. 2005 Nov;6(11):1076-81. doi: 10.1038/sj.embor.7400514. Epub 2005 Sep 9. PMID: 16170310; PMCID: PMC1371024). We used the same H2O2 concentrations also in the experiment presented in figure 1F.
Point 4. In Figure 5A, the presence of a GFP-Atg8 dot usually represents the formation of a mature autophagosome. Analysis of phagophore assembly site (PAS) localization of GFP-Atg8 would require high-resolution microscopy and needs to be confirmed by co-localization with other PAS marker proteins like Atg1. Also, the vacuolar proximity of the GFP-Atg8 cannot be commented on without the use of a vacuolar dye and co-localization analysis of GFP-Atg8 with this dye.
Response 4. We are aware of this and we’re planning to perform these experiments in a future paper
Point 5. The representative Figure in Figure 5A doesn't reflect the statistics in Figure 5B. The authors should select more representative images.
Response 5. We selected a more representative image for figure 5A
Point 6. The description of the results for the PD phase in Figure 5C and its correlation with Figure 2C should be elaborately explained including the details of the strains that are being referred to.
Response 6. We rephrased these results taking into account the referee suggestions
While some of the minor comments are:
Point 7. The results of the Lsm4 RGG domain as explained in lines 51-54 are not contradictory. The cited articles and the text both indicate that Lsm4 RGG is involved but not essential for histone H2A degradation. The authors could revise this.
Response 7. We modified the text accordingly
Point 8. The authors have referred to the proteins with and without the letter "p". The authors should maintain consistency in their use of nomenclature throughout the manuscript.
Response 8. We checked for these errors and maintained consistency referring to proteins without the letter “p”
Point 9. The description of the results for Figure 1G could be reworded as this assay has only been tried with caffeine with K. lactis. Also, related studies could be cited here again.
Response 9. We added the reference for the caffeine treatment (Mazzoni C, Mancini P, Madeo F, Palermo V, Falcone C. A Kluyveromyces lactis mutant in the essential gene KlLSM4 shows phenotypic markers of apoptosis. FEMS Yeast Res. 2003 Oct;4(1):29-35. doi: 10.1016/S1567-1356(03)00151-X. PMID: 14554194.)
Point 10. Figures 1A and 1C need to improve in quality, especially for the channels depicting DAPI and DHR123 staining respectively.
Response 10. We improved the quality improving the resolution of the pictures
Point 11. Since Figures 2 and 3 both describe autophagy-related assays, they could be included in a single Figure.
Response 11. As suggested by the reviewer 1, we combined figure 2 and 3 into a single figure
Point 12. The results on oxidative stress should be made a separate sub-section in the results.
Response 12. As suggested by the reviewer 1, we added a separate sub-session for oxidative stress results
Point 13. The reference Figure for the description on lines 251-252 and 256-258 is Figure 5A and not 5B.
Response 13. We modified the text accordingly
Point 14. The Y-axis title of Figures 5B, 5D, and 5F should be renamed to "% of cells with GFP-Atg8 dots". Similarly, the description of the results should refer to the percentages as the number of cells with GFP-Atg8 dots
Response 14. We modified the text accordingly
Point 15. authors could mention that the group in Figure 5B with -N is with 4h of treatment. Similarly, the group in 5D is with 16h of -N treatment. It would help to better appreciate their results across the entire Figure.
Response 15. We modified the text accordingly
Reviewer 2 Report
This paper is an interesting and self sufficient part of ongoing research on function of LSM genes. Here are remarks and questions:
24 - LSM4 should be italic type
Table 1 - also italic type for genes in two last lines, “MAT” instead of “Mat”, “LEU-GAL1-SDB23” – its better to use one alias (name) for LSM4 gene.
51 – “Similarly to yeast, this region is important for …” - maybe it will be better to say “Both in human and yeast cells …” or something like that
102 - chronological Life Span (CLS) - Chronological Life Span (CLS)
123 – “grow in galactose” - maybe it will be better to say “grow on galactose”
129-137 - for described experiments it should be explained, how exponential and stationary phases were determined. Samples were taken after same cultivation times for both wild type and Sclsm4Δ1 strains? Or these strains demonstrated such significant differences in growth rates that samples were taken at different times? It would be ideal to demonstrate and compare growth curves for these strains. The whole procedure for these experiments should be better described in “Materials and methods” – what volumes of which media (YPD or SD?) were used, how they were inoculated with starting culture. Here and further it would also be helpful to first mention exactly which strains were used in the experiment and only after that refer to them in shortened names (wild type and Sclsm4Δ1 strains).
159 Figure 1 – “***p-value<0.001, ****p-value><0.0001” – here and further (or separately in “Materials and methods” section) it should be mentioned with which statistical criteria data was analyzed.
166 - YPY medium containing 2% glycerol should be also described in “Materials and methods” along with YPD and SD.
182-185 “Protein extracts obtained from cells in the post-diauxic phase (PD, lanes 2 and 5) and nitrogen starvation (SD-N, lanes 3 and 6) showed that free GFP production was reduced in Sclsm4D1 cells, suggesting a defect in this mutant in inducing macroautophagy.” In figure 2B results supporting these claims are presented. However, only for nitrogen deprived medium (SD-N) statistical significance is demonstrated (*p- 192 value<0.05). Were results statistically significant for post-diauxic phase? Procedure for analysis of WB pictures may be described in brief in “Materials and methods”. What software was used?
174 – Maybe between “… a GFP-Atg8 processing assay [35]” and “As GFP β-barrel structure is 174 more resistant than …” there should be some phrase that wild type and Sclsm4Δ1 strains were transformed with pUG36/ATG8 resulting in synthesis of chimerical protein GFP-Atg8. Or mention the corresponding strains in table 1 (e.g. MCY4/Sclsm4Δ1 pUG36/ATG8).
As I understand in pUG36/ATG8 plasmid chimerical gene is under control of the promoter of MET17 gene that is involved in the biosynthesis of the sulfur-containing amino acids. Defective autophagy cells fail to maintain physiological levels of amino acids. Thus, in such cells activity of the genes involved in biosynthesis of amino acids may be significantly different from wild type. Maybe even during exponential phase. Was synthesis of chimerical protein GFP-Atg8 in wild type strain and in Sclsm4Δ1 strain during exponential phase at the same levels? Or it was higher in Sclsm4Δ1 strain in all replica measurements? Can this impact the results of GFP-Atg8 processing assay? This should be discussed.
249 “Sclsm4 1” - Sclsm4Δ1
pRS313/Sclsm4Δ1 and pRS313/ScLSM4 plasmids have same native promoter of LSM4 gene and full coding sequence or only part of it. This results in synthesis of full protein in MCY4/ScLSM4 strain and shortened protein in MCY4/Sclsm4Δ1 strain. Different effects of such proteins are then analyzed and compared in the paper. But according to the part of the paper describing plasmid construction and primers presented in Table 2 there is another important difference in these plasmids. pRS313/Sclsm4Δ1 contain only native promoter and part of LSM4 gene. While pRS313/ScLSM4 also contains 144 bp sequence corresponding to 3’UTR region of LSM4 gene. As we know 3’end processing of transcribed mRNAs and 3’UTR region of mature mRNAs significantly influence gene expression. Myself I have encountered instances when deletion of 3’ region in yeast vectors lead to more than 3 time decrease in protein production. Correct choice not only of promoter but also of terminator region is an important step of vector design for protein synthesis in biotechnology. This means that in experiments You have two variables – truncation of the protein and possible different expression levels because of different mRNA stability and processing caused by presence or absence of LSM4 gene 3’ region in plasmids. It may be proposed that “The Sclsm4Δ1 mutant of S. cerevisiae, which expresses a truncated form of the essential gene LSM4, showed premature ageing, fragmented nuclei and ROS accumulation” (290-290) not because of truncation of the protein, but because of different expression levels of LSM4 gene variants on the plasmids. This problem must be addressed. Either present additional experimental data, e.g. RQ-PCR results demonstrating similar mRNA levels for LSM4 variants in yeast cells. Or present comprehensive discussion with evidence proving that observed differences are indeed caused by truncation of the protein and not by the differences in structure of expression cassettes.
299 “autophagy [44]: our results” - why “:”?
Figure 1G – very interesting results are presented demonstrating different influence of deletion (Sclsm4D1) in dependence on type of carbon source. On YPD medium with dextrose wild type and Sclsm4Δ1 strains demonstrate small differences in growth. On YPY medium with glycerol Sclsm4Δ1 growth of strain is impaired. In discussion there is only some words regarding mitochondria and respiratory carbon sources (317-321). These results should be discussed more. Why effect of the deletion is more serious on glycerol?
A conclusion would be a good addition to this paper. Maybe move part of introduction (106-114) with main results of the work here?
Author Response
We thank the reviewer for her/his constructive criticisms, which allowed us to improve our work. In the new manuscript, the changed parts are highlighted in yellow.
Below are the point-by-point answers to the reviewer’s criticisms.
Point 1. 24 - LSM4 should be italic type
Response 1. We modified the text accordingly
Point 2. Table 1 - also italic type for genes in two last lines, “MAT” instead of “Mat”, “LEU-GAL1-SDB23” – its better to use one alias (name) for LSM4 gene.
Response 2. We modified the text accordingly
Point 3. 51 – “Similarly to yeast, this region is important for …” - maybe it will be better to say “Both in human and yeast cells …” or something like that
Response 3. We modified the text accordingly
Point 4. 102 - chronological Life Span (CLS) - Chronological Life Span (CLS)
Response 4. We modified the text accordingly
Point 5 123 – “grow in galactose” - maybe it will be better to say “grow on galactose”
Response 5. We modified the text accordingly
Point 6. 129-137 - for described experiments it should be explained, how exponential and stationary phases were determined. Samples were taken after same cultivation times for both wild type and Sclsm4Δ1 strains? Or these strains demonstrated such significant differences in growth rates that samples were taken at different times? It would be ideal to demonstrate and compare growth curves for these strains.
Response 6. We added a growth curve for both wild type and Sclsm4Δ1 strains as supplemental material and we indicated the specific growth rate.
Point 7. The whole procedure for these experiments should be better described in “Materials and methods” – what volumes of which media (YPD or SD?) were used, how they were inoculated with starting culture. Here and further it would also be helpful to first mention exactly which strains were used in the experiment and only after that refer to them in shortened names (wild type and Sclsm4Δ1 strains).
Response 7. We integrated the “Materials and Methods” section with the required description.
Point 8. 159 Figure 1 – “***p-value<0.001, ****p-value><0.0001” – here and further (or separately in “Materials and methods” section) it should be mentioned with which statistical criteria data was analyzed.
Response 8. A section (5.9) regarding statistical analysis has been added in “Materials and methods”
Point 9. 166 - YPY medium containing 2% glycerol should be also described in “Materials and methods” along with YPD and SD.
Response 9. YPY description has been added in “Materials and methods”
Point 10. 182-185 “Protein extracts obtained from cells in the post-diauxic phase (PD, lanes 2 and 5) and nitrogen starvation (SD-N, lanes 3 and 6) showed that free GFP production was reduced in Sclsm4D1 cells, suggesting a defect in this mutant in inducing macroautophagy.” In figure 2B results supporting these claims are presented. However, only for nitrogen deprived medium (SD-N) statistical significance is demonstrated (*p- 192 value<0.05). Were results statistically significant for post-diauxic phase? Procedure for analysis of WB pictures may be described in brief in “Materials and methods”. What software was used?
Response 10. Softwares used for WB were added in “Materials and methods”
Point 11. 174 – Maybe between “… a GFP-Atg8 processing assay [35]” and “As GFP β-barrel structure is 174 more resistant than …” there should be some phrase that wild type and Sclsm4Δ1 strains were transformed with pUG36/ATG8 resulting in synthesis of chimerical protein GFP-Atg8. Or mention the corresponding strains in table 1 (e.g. MCY4/Sclsm4Δ1 pUG36/ATG8).
Response 11. We modified the text accordingly
Point 12. As I understand in pUG36/ATG8 plasmid chimerical gene is under control of the promoter of MET17 gene that is involved in the biosynthesis of the sulfur-containing amino acids. Defective autophagy cells fail to maintain physiological levels of amino acids. Thus, in such cells activity of the genes involved in biosynthesis of amino acids may be significantly different from wild type. Maybe even during exponential phase. Was synthesis of chimerical protein GFP-Atg8 in wild type strain and in Sclsm4Δ1 strain during exponential phase at the same levels? Or it was higher in Sclsm4Δ1 strain in all replica measurements? Can this impact the results of GFP-Atg8 processing assay? This should be discussed.
Response 12. Actually, the fusion protein GFP-Atg8 accumulates at higher level in the mutant both in exponential and post-diauxic phase, suggesting a stabilization at mRNA or protein level that need to be investigated in the future. We discussed the higher level of GFP-Atg8 in the text
Point 13. 249 “Sclsm4 1” - Sclsm4Δ1
Response 13. We modified the text accordingly
Point 14. pRS313/Sclsm4Δ1 and pRS313/ScLSM4 plasmids have same native promoter of LSM4 gene and full coding sequence or only part of it. This results in synthesis of full protein in MCY4/ScLSM4 strain and shortened protein in MCY4/Sclsm4Δ1 strain. Different effects of such proteins are then analyzed and compared in the paper. But according to the part of the paper describing plasmid construction and primers presented in Table 2 there is another important difference in these plasmids. pRS313/Sclsm4Δ1 contain only native promoter and part of LSM4 gene. While pRS313/ScLSM4 also contains 144 bp sequence corresponding to 3’UTR region of LSM4 gene. As we know 3’end processing of transcribed mRNAs and 3’UTR region of mature mRNAs significantly influence gene expression. Myself I have encountered instances when deletion of 3’ region in yeast vectors lead to more than 3 time decrease in protein production. Correct choice not only of promoter but also of terminator region is an important step of vector design for protein synthesis in biotechnology. This means that in experiments You have two variables – truncation of the protein and possible different expression levels because of different mRNA stability and processing caused by presence or absence of LSM4 gene 3’ region in plasmids. It may be proposed that “The Sclsm4Δ1 mutant of S. cerevisiae, which expresses a truncated form of the essential gene LSM4, showed premature ageing, fragmented nuclei and ROS accumulation” (290-290) not because of truncation of the protein, but because of different expression levels of LSM4 gene variants on the plasmids. This problem must be addressed. Either present additional experimental data, e.g. RQ-PCR results demonstrating similar mRNA levels for LSM4 variants in yeast cells. Or present comprehensive discussion with evidence proving that observed differences are indeed caused by truncation of the protein and not by the differences in structure of expression cassettes.
Response 14. We added the LSM4 expression analysis by RQ-PCR in CML39-11A, MCY4/ScLSM4 and MCY4/Sclsm4Δ1. This analysis showed that Sclsm4Δ1 is highly expressed, eliminating the possibility that the premature ageing, fragmented nuclei and ROS accumulation are due a decrease in gene expression.
Point 15. 299 “autophagy [44]: our results” - why “:”?
Response 15. We modified the text accordingly
Point 16. Figure 1G – very interesting results are presented demonstrating different influence of deletion (Sclsm4D1) in dependence on type of carbon source. On YPD medium with dextrose wild type and Sclsm4Δ1 strains demonstrate small differences in growth. On YPY medium with glycerol Sclsm4Δ1 growth of strain is impaired. In discussion there is only some words regarding mitochondria and respiratory carbon sources (317-321). These results should be discussed more. Why effect of the deletion is more serious on glycerol?
Response 16. This is a very interesting question. We demonstrated that in Kllsm4Δ1 this could be linked to fragmented mitochondrial morphology and this could be reverted by Acetyl-L-carnitine supplementation or by PGK1, HIR1 or NEM1 overexpression. Moreover, from genome-wide studies it has been found that also lsm1D mutant cannot grow on glycerol, suggesting a probable link between mRNA metabolism and respiration. We added these considerations to the discussion.
Point 17. A conclusion would be a good addition to this paper. Maybe move part of introduction (106-114) with main results of the work here?
Response 17. We accepted the reviewer suggestion and moved the introduction part to the conclusion.
Round 2
Reviewer 1 Report
The authors Caraba et al., have submitted a revised version of their manuscript titled "Yeast lsm pro-apoptotic mutants show defects in autophagy induction". The authors have answered most of the queries raised in the previous round of review in this version of the manuscript. However, a number of concerns remain.
The authors do not have any evidence to show that the Lsm4 or Lsm1 mutant has defects in autophagy induction. Their results show that there is a block in autophagy. Therefore, I recommend dropping the word "induction" from the title.
The accumulation of GFP-Atg8 dots in Figure 4 is evidence of a block in vacuolar autophagosome internalization. Since the authors do not have any data regarding the effect of PAS, the authors should remove references to PAS from the abstract.
In the description for Figure 1, the authors should describe the results of the strain expressing the full form ScLSM4 at the same location as the description for the other groups.
The authors could rearrange the current Figure 2C as Figure 1F as both current Figure 1E and current Figure 2C check the effects of the deletion mutant on cell viability under different conditions. Next, since the current Figure 2E confirms known data about the involvement of Lsm1 (from the cited article) and Lsm4 (from the earlier figure) in autophagy, I would move it to supplementary. Finally, since current Figure 1F, current Figure 2D, and current Figure 3 discuss the effect of oxidative stress and Rapamycin, I would merge them as a single Figure.
If the autophagy pathway was affecting the activation of MET17, there would be an effect on the expression levels of GFP-Atg8. However, the expression of GFP-Atg8 is similar between WT and Sclsm4Δ1. Therefore, there is no effect on MET17.
The authors cite studies to justify the experiment in current Figure 3 to understand if Lsm4 could be playing a role in protection against aging-induced oxidative stress by regulating autophagy. However, since their assay results in killing the wild-type cells at the first concentration that they tested (0.8mM), the amount of oxidative stress induced by aging could be much lower than what is induced by 0.8mM H2O2. Therefore, the authors need to optimize their assay. Using a further lower concentration of H2O2 as described in Poljak A, Dawes IW, Ingelse BA, Duncan MW, Smythe GA, Grant CM. Oxidative damage to proteins in yeast cells exposed to adaptive levels of H(2)O(2). Redox Rep. 2003;8(6):371-7. doi: 10.1179/135100003225003401. PMID: 14980070. could help provide a window through which the authors could see the effect of rapamycin-induced autophagy on aging oxidative stress.
In Figure 4A, GFP-Atg8 dots are representative of autophagosomes and not PAS. As mentioned in the previous round of review, the authors can comment on structures like PAS only with the support of high-resolution microscopy. Since the authors plan to do those studies in the future, the authors should revise the text describing the dots in Figure 4A as autophagosomes.
The image panels in Figures 4C and 4E could be rearranged to have wild-type and mutant exposed to the same condition adjacent to each other. This will make the comparisons easier for the reader.
In the description of Figure 4C, the authors could also mention that at 16h of SD-N, the wild-type cells had GFP in the vacuole, indicating active autophagy flux while the mutant did not have any GFP in the vacuole, indicating a block in autophagy.
The authors should revise incorrect references of Figure 4E and 4F as 5E and 5F respectively in the text. Also, In Figure 4D, the authors have incorrectly labeled PD as SD and similarly, the authors should also revise SD to SD stat in Figure 4F.
The results of the GFP-Atg8 localization assay with the Lsm1 mutant have significant differences from the results from the Lsm4 mutant especially concerning the number of cells with 2 or more GFP-Atg8 dots. Such nuances should be mentioned and their biological significance described. Interestingly, the authors observe a drop in the percentage of wild-type cells with GFP-Atg8 dots when exposed to SD-N for 16h in the supplementary Figure 3 while the wild-type strain used in Figure 4C shows an increase in the percentage of cells. Such differences could be due to the use of different backgrounds of wild-type strains and should be mentioned in the results.
The authors have used a different magnification for the images in Figure 4 and Supplementary Figure 3. The authors should make them uniform.
English editing is required for a better understanding of the result and increased appreciation of the overall manuscript.
Author Response
We thank the reviewer for her/his constructive criticisms, which allowed us to improve our work. In the new manuscript, the changed parts are highlighted in blue.
Below are the point-by-point answers to the reviewer’s criticisms.
The authors Caraba et al., have submitted a revised version of their manuscript titled "Yeast lsm pro-apoptotic mutants show defects in autophagy induction". The authors have answered most of the queries raised in the previous round of review in this version of the manuscript. However, a number of concerns remain.
Point1. The authors do not have any evidence to show that the Lsm4 or Lsm1 mutant has defects in autophagy induction. Their results show that there is a block in autophagy. Therefore, I recommend dropping the word "induction" from the title.
Response 1. We removed the word "induction" from the title.
Point 2. The accumulation of GFP-Atg8 dots in Figure 4 is evidence of a block in vacuolar autophagosome internalization. Since the authors do not have any data regarding the effect of PAS, the authors should remove references to PAS from the abstract.
Response 2. We removed references to PAS from the abstract
Point 3. In the description for Figure 1, the authors should describe the results of the strain expressing the full form ScLSM4 at the same location as the description for the other groups.
Response 3: Accordingly with the referee request, we described in the text the results of the strain expressing the full form ScLSM4 at the same location as the description for the other groups.
Point 4. The authors could rearrange the current Figure 2C as Figure 1F as both current Figure 1E and current Figure 2C check the effects of the deletion mutant on cell viability under different conditions. Next, since the current Figure 2E confirms known data about the involvement of Lsm1 (from the cited article) and Lsm4 (from the earlier figure) in autophagy, I would move it to supplementary. Finally, since current Figure 1F, current Figure 2D, and current Figure 3 discuss the effect of oxidative stress and Rapamycin, I would merge them as a single Figure.
Response 4. As required by the referee, we rearranged the Figure 2C as Figure 1F. We moved the current figure 2E to supplementary and merged Figure 2D and Figure 3 in the new figure 3
Point 5. If the autophagy pathway was affecting the activation of MET17, there would be an effect on the expression levels of GFP-Atg8. However, the expression of GFP-Atg8 is similar between WT and Sclsm4Δ1. Therefore, there is no effect on MET17. The authors cite studies to justify the experiment in current Figure 3 to understand if Lsm4 could be playing a role in protection against aging-induced oxidative stress by regulating autophagy. However, since their assay results in killing the wild-type cells at the first concentration that they tested (0.8mM), the amount of oxidative stress induced by aging could be much lower than what is induced by 0.8mM H2O2. Therefore, the authors need to optimize their assay. Using a further lower concentration of H2O2 as described in Poljak A, Dawes IW, Ingelse BA, Duncan MW, Smythe GA, Grant CM. Oxidative damage to proteins in yeast cells exposed to adaptive levels of H(2)O(2). Redox Rep. 2003;8(6):371-7. doi: 10.1179/135100003225003401. PMID: 14980070. could help provide a window through which the authors could see the effect of rapamycin-induced autophagy on aging oxidative stress.
Response 5. As required by the referee, we performed the oxidative stress with lower amount of H2O2, but also with these concentrations, we could not observe any difference between strains treated with rapamycin compared with the untreated ones. These results were obtained with both a drop-test assay and a more precise viability test assay, We reported these experiment in the new Supplementary figure 5.
Point 6. In Figure 4A, GFP-Atg8 dots are representative of autophagosomes and not PAS. As mentioned in the previous round of review, the authors can comment on structures like PAS only with the support of high-resolution microscopy. Since the authors plan to do those studies in the future, the authors should revise the text describing the dots in Figure 4A as autophagosomes.
Response 6. We revised the text accordingly.
Point 7. The image panels in Figures 4C and 4E could be rearranged to have wild-type and mutant exposed to the same condition adjacent to each other. This will make the comparisons easier for the reader. In the description of Figure 4C, the authors could also mention that at 16h of SD-N, the wild-type cells had GFP in the vacuole, indicating active autophagy flux while the mutant did not have any GFP in the vacuole, indicating a block in autophagy.
Response 6. We revised the text accordingly.
Point 7. The authors should revise incorrect references of Figure 4E and 4F as 5E and 5F respectively in the text. Also, In Figure 4D, the authors have incorrectly labeled PD as SD and similarly, the authors should also revise SD to SD stat in Figure 4F.
Response 7. We revised the Figure accordingly.
Point 8. The results of the GFP-Atg8 localization assay with the Lsm1 mutant have significant differences from the results from the Lsm4 mutant especially concerning the number of cells with 2 or more GFP-Atg8 dots. Such nuances should be mentioned and their biological significance described. Interestingly, the authors observe a drop in the percentage of wild-type cells with GFP-Atg8 dots when exposed to SD-N for 16h in the supplementary Figure 3 while the wild-type strain used in Figure 4C shows an increase in the percentage of cells. Such differences could be due to the use of different backgrounds of wild-type strains and should be mentioned in the results.
Response 8. We revised the text accordingly.
Point 9. The authors have used a different magnification for the images in Figure 4 and Supplementary Figure 3. The authors should make them uniform.
Response 9. The magnification of Figure 4 and Figure S3 should be the same
Comments on the Quality of English Language
Point 10 English editing is required for a better understanding of the result and increased appreciation of the overall manuscript.
Response 10. We extensively edited the text for English, we hope that this effort will ameliorate the understanding of the overall manuscript
Reviewer 2 Report
I acept all corrections and answers to my questions. I only found some minor mistakes in new version of the paper:
"although the growth is slightly slower than the wild strain and the one expressing the full form of the LSM4 gene (supplementary Figure 1) ." - an extra space before dot.
"The cells were counted manually with ImageJ software [59]." - reference [61] but not [59] corresponds to ImageJ software. You should carefully check other references that may be impakted by that.
Author Response
We thank the reviewer for her/his constructive criticisms, which allowed us to improve our work. In the new manuscript, the changed parts are highlighted in blue.
Below are the point-by-point answers to the reviewer’s criticisms.
I acept all corrections and answers to my questions. I only found some minor mistakes in new version of the paper:
Point 1. "although the growth is slightly slower than the wild strain and the one expressing the full form of the LSM4 gene (supplementary Figure 1) ." - an extra space before dot.
Response 1. We revised the text accordingly.
Point 2 "The cells were counted manually with ImageJ software [59]." - reference [61] but not [59] corresponds to ImageJ software. You should carefully check other references that may be impakted by that.
Response 2. We corrected this mistake.